# Synthesis of Novel Triazinoindole-Based Thiourea Hybrid: A Study on α-Glucosidase Inhibitors and Their Molecular Docking

**DOI:** 10.3390/molecules24213819

**Published:** 2019-10-23

**Authors:** Muhammad Taha, Foziah J. Alshamrani, Fazal Rahim, Shawkat Hayat, Hayat Ullah, Khalid Zaman, Syahrul Imran, Khalid Mohammed Khan, Farzana Naz

**Affiliations:** 1Department of Clinical Pharmacy, Institute for Research and Medical Consultations (IRMC), Imam Abdulrahman Bin Faisal University, P.O. Box 1982, Dammam 31441, Saudi Arabia; 2Neurology Department, King Fahad hospital of University, Imam Abdulrahman Bin Faisal University, P.O. Box 1982, Dammam 34211, Saudi Arabia; Fshamrani@uod.edu.sa; 3Department of Neuroscience Research, Institute for Research and Medical Consultations (IRMC), Imam Abdulrahman Bin Faisal University, P.O. Box 1982, Dammam 31441, Saudi Arabia; 4Department of Chemistry, Hazara University, Mansehra-21300, Khyber Pakhtunkhwa, Pakistan; shawkathayat866@yahoo.com (S.H.); ayaanwazir366@gmail.com (H.U.); khalidchemist69@yahoo.com (K.Z.); 5Atta-ur-Rahman Institute for Natural Products Discovery (AuRIns), Universiti Teknologi MARA Puncak Alam Campus, 42300 Bandar Puncak Alam, Selangor D.E., Malaysia; imran@isiswa.uitm.edu.my; 6Faculty of Applied Sciences, Universiti Teknologi MARA, 40450 Shah Alam, Selangor D.E., Malaysia; 7H. E. J. Research Institute of Chemistry, International Center for Chemical and Biological Sciences, University of Karachi, Karachi 75270, Pakistan; khalid.khan@iccs.edu; 8Department of Chemistry, Jinnah University for Women, 5-C, Nazimabad, Karachi 74600, Pakistan; farzanahej@yahoo.com

**Keywords:** synthesis, triazinoindole, thiosemicarbazide, alpha-glucosidase, molecular docking study, SAR

## Abstract

A new class of triazinoindole-bearing thiosemicarbazides (**1**–**25**) was synthesized and evaluated for α-glucosidase inhibitory potential. All synthesized analogs exhibited excellent inhibitory potential, with IC_50_ values ranging from 1.30 ± 0.01 to 35.80 ± 0.80 μM when compared to standard acarbose (an IC_50_ value of 38.60 ± 0.20 μM). Among the series, analogs **1** and **23** were found to be the most potent, with IC_50_ values of 1.30 ± 0.05 and 1.30 ± 0.01 μM, respectively. The structure–activity relationship (SAR) was mainly based upon bringing about different substituents on the phenyl rings. To confirm the binding interactions, a molecular docking study was performed.

## 1. Introduction

Diabetes mellitus is a chronic health-threatening metabolic disease that is caused by insufficient insulin secretion and is categorized as hypoglycemia/hyperglycemia [1]. In type II diabetes mellitus, enhanced postprandial glucose levels can increase the risk of developing stroke, atherosclerosis, and other coronary diseases [2]. In order to treat type II diabetes and its complications, the inhibition of digestive enzyme-like α-glucosidase is an effective approach that can reduce postprandial glucose and risk factors [3]: α-glucosidase is located in the epithelium cell lining of the small intestine and is responsible for the conversion of polysaccharides and disaccharides into glucose. The inhibition of α-glucosidase is directly associated with the blood glucose level, and its inhibition is vital due to the potential effect of a decrease in postprandial blood glucose levels [4]. In order to delay rapid blood glucose production, certain α-glucosidase inhibitors such as acarbose and voglibose are used clinically, though there are certain side effect that include abdominal pain, diarrhea, and other gastrointestinal disorders in chronic therapy [5]. Therefore, in order to treat postprandial hyperglycemia, a search for efficient and safe α-glucosidase inhibitors is needed.

Triazinoindole scaffolds possess excellent biological potential against malarial and viral diseases. Substituted triazinoindole scaffolds are of considerable interest due to their excellent antihypertensive [6], antidepressant [7], anti-inflammatory [8], antihypoxic [9], antifungal, and antibacterial activities [10]. Selected triazinoindole compounds act as potential drugs in treating the common cold [11,12,13,14].

Our research group has been working on the design and synthesis of heterocyclic compounds in search of potential lead compounds for many years, and we have found promising results [15,16,17,18,19,20,21,22,23,24,25,26,27,28,29]. We have already reported on triazinoindole analogs as potent α-glucosidase inhibitors [30]. Thus, we decided to screen a library of triazinoindole-bearing thiosemicarbazide analogs for α-glucosidase activity (Figure 1).

## 2. Results and Discussion

### 2.1. Chemistry

A new class of triazinoindole-based thiosemicarbazide analogs (**1**–**25**) was carried out in three steps.

In the first step, thiosemicarbazide was reacted and refluxed with isatin in H_2_O in the presence of potassium carbonate to yield 5*H*-triazinoindole-3-thiol as an intermediate product (**I**). The intermediate (**I**) was then mixed and refluxed with different substituted phenacyl bromides in EtOH in the presence of Et_3_N to give triazinoindole derivatives as a second intermediate product (**II**). (Scheme 1)

In the second step, hydrazine hydrate was reacted and refluxed with different isothiocyanates in methanol to yield a thiosemicarbazide derivative as an intermediate (**III**). (Scheme 2)

In the third step, intermediate product (**II**) was reacted and refluxed with intermediate product (**III**) in glacial acetic acid to give the final product, triazinoindole-bearing thiosemicarbazide (**1**–**25**). (Scheme 3, Table 1).

### 2.2. Biological Activity

A new class of triazinoindole-bearing thiosemicarbazide analogs (**1**–**25**) was synthesized and was evaluated for α-glucosidase inhibitory potential. All of the synthesized scaffolds exhibited outstanding inhibitory potential, with IC_50_ values ranging from 1.30 ± 0.01 to 35.80 ± 0.80 μM when compared to standard acarbose with an IC_50_ value of 38.60 ± 0.20 μM. The structure–activity relationship (SAR) was mainly based upon different substation pattern on phenyl rings.

We compared compound **1** (IC_50_ = 1.30 ± 0.05 μM) with a methoxy moiety at position 4 on one phenyl ring and two chloro groups at positions 2 and 3 on the second phenyl ring to scaffold **18** (IC_50_ = 2.30 ± 0.05 μM), which also had a methoxy moiety at position 4 on one phenyl ring and two chloro groups at positions 3 and 4 on the second phenyl ring. The inhibition difference in these two scaffolds may have been due to two chloro groups that were present in different positions on the second phenyl ring.

If we compare scaffold **9** (IC_50_ value 6.80 ± 0.10 μM) to scaffold **17** (IC_50_ value 8.80 ± 0.20 μM), both scaffolds had a nitro moiety on one phenyl ring, and in both cases the other phenyl ring was unsubstituted. In scaffold **9**, the nitro moiety was present at position 4, while in scaffold **17**, the nitro moiety was present at position 3 on the phenyl ring. The potential difference in these two scaffolds may have been due to the nitro moiety being in a different position on one phenyl ring.

Similarly, we compared compound **15** (IC_50_ = 5.80 ± 0.20 μM) with a methoxy moiety at position 4 on one phenyl ring and two methyl groups at positions 2 and 6 on the second phenyl ring to scaffold **22** (IC_50_ = 5.90 ± 0.10 μM), which also had a methoxy moiety at position 4 on one phenyl ring and two methyl groups at positions 2 and 3 on the second phenyl ring. The inhibition difference in these two scaffolds may have been due to two methyl groups that were present in different positions on the second phenyl ring.

It was observed over the whole study that the phenyl ring substituents’ nature as well as their positions greatly affected the inhibitory potential of the compound. A docking study was done to understand the binding interactions of the most active scaffolds with the enzyme active site.

### 2.3. Docking Study

Docking studies were carried out on scaffolds **1**, **16**, and **23**, which displayed the most potent inhibitory potential among the whole series. Prior to docking and analysis of the binding mode of the most active compound, **1**, the docking method was validated through the control docking of a native inhibitor. Acarbose was docked into α-glucosidase from sugar beet (PDB code: 3W37) and was compared by superimposing the native ligand in the protein, as mentioned in Imran et. al. (2016) [24]. Prior to docking and analysis of the binding mode of the active compounds, the docking method was validated through the control docking of a native inhibitor. Acarbose was docked into α-glucosidase from sugar beet (PDB code: 3W37) and was compared by superimposing the native ligand in the protein (Figure 2a). The rmsd value between the docked and actual pose of acarbose was found to be 0.65 Å. Even though α-glucosidase from sugar beet shared a relatively low homology with Baker’s yeast α-glucosidase (16% identity), the active site was highly conserved, and the main interactions of the ligand remained the same. Another control docking was done on the target protein, isomaltase from Baker’s yeast, using α-d-glucopyranose, which was the native ligand located in the active site (Figure 2b). The rmsd value for the docked pose and the native ligand was found to be 0.93 Å.

The Chemscore values of the active compounds are reported in Table 2, and these values correlated well with the IC_50_ values. The results obtained showed that scaffold **1** recorded the highest Chemscore value, −89.3 kJ/mol, followed by compound **23**, which recorded a Chemscore value of −87.7 kJ/mol, and finally compound **16**, which displayed the lowest Chemscore value, −74.5.

The results from the docking of these active compounds showed that they were able to form several hydrogen bonds within the cavity. The hydrogen on the nitrogen of triazinoindole formed a hydrogen bond with the backbone (Oε2) of Glu276, and the catalytic residue was involved in the hydrolysis reaction at a distance of 2.16 Å.

In the case of compound **1**, one of the nitrogens on the triazine moiety established a hydrogen bonding interaction with the residue of hydrophobic patch Phe300 at a distance of 3.42 Å. The sulfur linkage displayed hydrogen bonding with the side chain (O) of Glu304 at a distance of 2.23 Å. An interaction involving a halide bond was observed between a chlorine substituent at the *meta* position and the side chain (O) of Thr307 at a distance of 1.89 Å. An aromatic ring containing a methoxy substituent formed an electrostatic π-hydrogen interaction with His239 (Hε1) at a distance of 2.64 Å. As for the triazine moiety, an electrostatic π-hydrogen involving the residue of hydrophobic patch Phe177 was expected to stabilize the ligand–enzyme complex alongside Tyr71, which formed an electrostatic π-hydrogen interaction with one of the hydrogens from the triazinoindole moiety at a distance of 3.93 Å Figure 3.

The analog **23** docking study revealed that this scaffold was capable of forming several hydrogen bonds within the cavity. The residue Glu276, through its backbone (Oε2), participated in hydrogen bonding interactions with the amino group of triazinoindole compound **23** at a distance of 2.43 Å. As for the triazine moiety, electrostatic π-hydrogen was involved in the side chain of residue Phe157, which was expected to stabilize the ligand–enzyme complex. Another hydrogen formed an electrostatic π-hydrogen interaction with Tyr71. It was also observed that His239 was capable of forming an electrostatic interaction with the oxygen of the nitro substituent at the *meta* position. An aromatic ring consisting of dichloro substituents were stabilized through electrostatic interactions with Pro309 through electrostatic π-hydrogen interactions (Figure 4).

The docking results for compound **16** displayed fewer interactions compared to compounds **1** and **23**. Some of the interactions that remained the same were the interactions of hydrogen on the nitrogen of triazinoindole, which formed a hydrogen bond with the backbone (Oε2) of Glu276 at a distance of 1.84 Å, and hydrogen formed electrostatic π-hydrogen interactions with Tyr71 at a distance of 3.98 Å. On the other hand, an aromatic ring consisting of a fluoro substituent at the *para* position displayed an electrostatic interaction with Glu304. It was observed that the CH_3_ of methoxy at the *para* position of the other aromatic ring displayed π-hydrogen interactions with Phe157 at a distance of 4.12 Å. (Figure 5)

## 3. Conclusions

In conclusion, we synthesized 25 analogs of triazinoindole-bearing thiosemicarbazide and evaluated them against the α-glucosidase enzyme. All of the synthesized scaffolds exhibited outstanding inhibitory potential, with IC_50_ values ranging from 1.30 ± 0.01 to 35.80 ± 0.80 μM when compared to standard acarbose with an IC_50_ value of 38.60 ± 0.20 μM. It was confirmed through the SAR that polar- and electron-withdrawing groups on the phenyl rings had a lot of influence on the potency of the compounds. A docking study was done to understand the binding interactions of the most active scaffolds.

## 4. Experiment

### 4.1. General Method for the Synthesis of Triazinoindole-Bearing Thiosemicarbazide Analogs (**1**–**25**)

A new class of triazinoindole-bearing thiosemicarbazide analogs (**1**–**25**) was carried out in three steps.

In the first step, thiosemicarbazide (10 mmol) was reacted and refluxed with isatin (10 mmol) in H_2_O in the presence of potassium carbonate (5 mmol) to yield 5*H*-triazinoindole-3-thiol as intermediate (**I**). Intermediate product (**I**) (5 mmol) was then mixed and refluxed with different substituted phenacyl bromides (5 mmol) in EtOH in the presence of Et_3_N to give a triazinoindole derivative as the second intermediate product (**II**).

In the second step, hydrazine hydrate (2 mL) was reacted and refluxed with different isothiocyanates (1 mmol) in methanol to yield a thiosemicarbazide derivative as intermediate product (**III**).

In the third step, the intermediate (**II**) was reacted and refluxed with an equimolar intermediate (**III**) in glacial acetic acid to give the final product, triazinoindole-bearing thiosemicarbazide.

*2-(2-((5H-[1,2,4]triazino [5,6-b]indol-3-yl)thio)-1-(4-methoxyphenyl)ethylidene)-N-(2,3-dichlorophenyl)hydrazine-1-carbothioamide* (**1**) Yield: 62%; ^1^H-NMR: (500 MHz, DMSO-*d*_6_), *δ* 13.30 (s, 1H, NH), 11.25 (s, 2H, NH), 7.84 (d, *J* = 7.3 Hz, 2H, Ar), 7.54 (d, *J* = 6.2 Hz, 1H, Ar), 7.45 (s, 1H, Ar), 7.35 (t, *J* = 5 Hz, 2H, Ar), 7.26 (dd, *J* = 1, 6.65 Hz, 1H, Ar), 7.10 (m, 4H, Ar), 5.2 (s, 2H, CH_2_), 3.81 (s, 3H, OCH_3_). ^13^C-NMR (125 MHz, DMSO-*d*_6_): *δ* 171.9, 165.8, 163.1, 159, 150.9, 141.2, 131.8, 131.8, 130.3, 128.2, 127.0, 126.9, 126.7, 123, 122.3, 121.2, 119.7, 119.7, 118.3 114.1, 114.1, 111, 104, 55.2, 41.2. HREI-MS: *m*/*z*, calcd for C_25_H_19_Cl_2_N_7_OS_2_ [M]^+^ 567.0470; found: 567.0458.

*2-(2-((5H-[1,2,4]triazino [5,6-b]indol-3-yl)thio)-1-(4-methoxyphenyl)ethylidene)-N-(4-nitrophenyl)hydrazine-1-carbothioamide* (**2**) Yield: 68%; ^1^H-NMR: (500 MHz, DMSO-*d*_6_), *δ* 11.25 (s, 1H, NH), 10.86 (s, 2H, NH), 8.24 (d, *J* = 7.3 Hz, 2H, Ar), 7.83 (d, *J* = 7.2 Hz, 2H, Ar), 7.79 (d, *J* = 7.65 Hz, 2H, Ar), 7.53 (d, *J* = 6.2 Hz, 1H, Ar), 7.34 (d, *J* = 1H, Ar), 7.10 (t, *J* = 6.3, 1H, Ar), 6.98 (t, *J* = 7.5, 3H, Ar), 4.91 (s, 2H, CH_2_), 3.81 (s, 3H, OCH_3_). ^13^C-NMR (125 MHz, DMSO-*d*_6_), *δ* 171.9, 165.8, 163.1, 159, 150.9, 141.2, 131.8, 131.8, 130.3, 128.2, 127, 126.9, 126.7, 123, 122.3, 121.2, 119.7, 119.7, 118.3 114.1, 114.1, 111, 104, 55.2, 38.2. HREI-MS: *m*/*z*, calcd for C_25_H_20_N_8_O_3_S_2_ [M]^+^ 544.1100; found: 544.1088.

*2-(2-((5H-[1,2,4]triazino [5,6-b]indol-3-yl)thio)-1-([1,1′-biphenyl]-4-yl)ethylidene)-N-(2,3-dichlorophenyl)hydrazine-1-carbothioamide* (**3**) Yield: 68%; 1H-NMR: (500 MHz, DMSO-*d*_6_), *δ* 11.25 (s, 1H, NH), 10.86 (s, 2H, NH), 8.24 (d, *J* = 7.3 Hz, 2H, Ar), 7.83 (d, *J* = 7.2 Hz, 2H, Ar), 7.79 (d, *J* = 7.65 Hz, 2H, Ar), 7.53 (d, *J* = 6.2 Hz, 1H, Ar), 7.34 (d, *J* = 6.5 Hz, 1H, Ar), 7.10 (t, *J* = 6.3 Hz, 1H, Ar), 6.98–6.93 (m, 3H, Ar), 4.91 (s, 2H, CH2), 3.81 (s, 3H, OCH_3_). ^13^C-NMR (125 MHz, DMSO-*d*_6_), *δ* 171.9, 165.8, 163.1, 159, 150.9, 141.2, 131.8, 131.8, 130.3, 128.2, 127, 126.9, 126.7, 123, 122.3, 121.2, 119.7, 119.7, 118.3 114.1, 114.1, 111, 104, 55.2, 38.2. HREI-MS: *m*/*z*, calcd for C_25_H_20_N_8_O_3_S_2_ [M]^+^ 544.1100; found: 544.1088.

*2-(2-((5H-[1,2,4]triazino[5,6-b]indol-3-yl)thio)-1-([1,1′-biphenyl]-4-yl)ethylidene)-N-(p-tolyl)hydrazine-1-carbothioamide* (**4**) Yield: 72%; ^1^H-NMR: (500 MHz, DMSO-*d*_6_), *δ* 11.25 (s, 2H, NH), 10.86 (s, 1H, NH), 8.2 (d, *J* = 7 Hz, 1H, Ar), 8.01 (m, 2H, Ar), 7.9 (d, *J* = 7 Hz, 1H, Ar), 7.80 (d, *J* = 6.2 Hz, 1H, Ar), 7.7 (m, 5H, Ar), 7.6 (d, *J* = 7.1 Hz, 2H, Ar), 7.4 (m, 3H, Ar), 7.1 (t, *J* = 6.25 Hz, 1H, Ar), 6.9 (d, *J* = 6.5 Hz, 1H, Ar), 4.9 (s, 2H, CH_2_), 3.0 (s, 3H, CH_3_). ^13^C-NMR (125 MHz, DMSO-*d*_6_), *δ* 192.8, 166.2, 163.1, 146.4, 144.8, 141, 139.5, 139.4, 138.8, 134.7, 133, 129.0, 129.0, 128.9, 128.4, 127.5, 127.0, 126.8, 126.8, 126.4, 126.2,122.4, 122.3, 121.4, 119.8, 117.5, 112.6, 111.0, 106.9, 38.2, 21.3. HREI-MS: *m*/*z*, calcd for C_31_H_25_N_7_S_2_ [M]^+^ 559.1613; found: 559.1601.

*2-(2-((5H-[1,2,4]triazino[5,6-b]indol-3-yl)thio)-1-(4-methoxyphenyl)ethylidene)-N-(p-tolyl)hydrazine-1-carbothioamide* (**5**) Yield: 73%; ^1^H-NMR: (500 MHz, DMSO-*d*_6_), *δ* 13.33 (s, 1H, NH), 11.25 (s,2H, NH), 7.8 (d, *J* = 7.25 Hz, 2H, Ar), 7.79 (d, *J* = 7.2 Hz, 2H, Ar), 7.54 (d, *J* = 6.2 Hz, 2H, Ar), 7.35 (m, 2H, Ar), 7.11 (t, *J* = 6.25 Hz, 2H, Ar), 7.02 (m, 2H, Ar), 5.2 (s, 2H, CH_2_), 3.80 (s, 3H, -OCH_3_), 2.42 (s, 3H, -CH_3_). ^13^C-NMR (125 MHz, DMSO-*d*_6_), *δ* 165.8, 163.1, 159.0, 150.9, 141.2, 131.8, 130.3, 127.0, 127.0, 127.0, 126.9, 126.7, 122.3, 122.3, 121.0, 119.7, 119.7,114.1, 114.1, 114.1, 114.0, 111.0,104.6, 55.2, 38.2, 22.0. HREI-MS: *m*/*z*, calcd for C_26_H_23_N_7_OS_2_ [M]^+^ 513.1405; found: 513.1390.

*2-(2-((5H-[1,2,4]triazino[5,6-b]indol-3-yl)thio)-1-([1,1′-biphenyl]-4-yl)ethylidene)-N-(2-bromophenyl)hydrazine-1-carbothioamide* (**6**) Yield: 77%; ^1^H-NMR: (500 MHz, DMSO-*d*_6_), *δ* 11.25 (s, 2H, NH), 10.86 (s, 1H, NH), 8.0 (d, *J* = 7 Hz, 1H, Ar), 7.9 (d, *J* = 7 Hz, 1H, Ar), 7.7 (d, *J* = 6.2 Hz, 1H, Ar), 7.5 (d, *J* = 7 Hz, 2H, Ar), 7.4 (d, *J* = 7.1 Hz, 2H, Ar), 7.36 (m, 3H, Ar), 7.34 (t, *J* = 6.25 Hz, 2H, Ar), 7.24 (d, *J* = 6.6 Hz, 1H, Ar), 7.21 (t, *J* = 7.3 Hz, 1H, Ar), 7.17 (d, *J* = 7 Hz, 1H, Ar), 7 (t, *J* = 6.9 Hz, 1H, Ar), 6.9 (d, *J* = 6.5 Hz, 1H, Ar), 4.9 (s, 2H, CH_2_). ^13^C-NMR (125 MHz, DMSO-*d*_6_), *δ* 192.8, 166.2, 163.1, 146.4, 144.8, 141, 139.5, 139.4, 138.8, 134.7, 133, 129.0, 129.0, 128.9, 128.4, 127.5, 127.0, 126.8, 126.8, 126.4, 126.2,122.4, 122.3, 121.4, 119.8, 117.5, 112.6, 111.0, 106.9, 38.2. HREI-MS: *m*/*z*, calcd for C_30_H_22_BrN_7_S_2_ [M]^+^ 623.0561; found: 623.0550.

*2-(2-((5H-[1,2,4]triazino[5,6-b]indol-3-yl)thio)-1-(p-tolyl)ethylidene)-N-(2-bromophenyl)hydrazine-1-carbothioamide* (**7**) Yield: 68%; ^1^H-NMR: (500 MHz, DMSO-*d*_6_), *δ* 12.5 (s, 2H, NH), 11.25 (s, 1H, NH), 8.28 (d, *J* = 6.45 Hz, 1H, Ar), 8.01 (d, *J* = 6.75 Hz, 3H, Ar), 7.69 (t, *J* = 6.65 Hz, 2H, Ar), 7.5 (d, *J* = 6.97 Hz, 2H, Ar), 7.43 (m, 4H, Ar), 4.8 (s, 2H, CH_2_), 2.42 (s, 3H, CH_3_). ^13^C-NMR (125 MHz, DMSO-*d*_6_), *δ* 171.9, 165.8, 163.1, 151.1, 141.2, 137.2, 131.9, 131.2, 130.4, 130.4, 130.4, 129.3, 129.2, 125.6, 125.6, 125.4, 122.3, 121.2, 119.7, 119.7, 111.0, 109.7, 105.8, 38.2, 20.7. HREI-MS: *m*/*z*, calcd for C_25_H_20_BrN_7_S_2_ [M]^+^ 561.0405; found: 561.0392.

*2-(2-((5H-[1,2,4]triazino[5,6-b]indol-3-yl)thio)-1-(p-tolyl)ethylidene)-N-(p-tolyl)hydrazine-1-carbothioamide* (**8**) Yield: 65%; ^1^H-NMR: (500 MHz, DMSO-*d*_6_): *δ* 12.5 (s, 2H, NH), 11.4 (s, 1H, NH), 8.25 (d, *J* = 6.45 Hz, 1H, Ar), 7.98 (d, *J* = 6.75 Hz, 3H, Ar), 7.69 (m, 2H, Ar), 7.48 (d, *J* = 6.97 Hz, 2H, Ar), 7.39 (m, 4H, Ar), 4.9 (s, 2H, CH_2_), 2.42 (s, 6H, CH_3_). ^13^C-NMR (125 MHz, DMSO-*d*_6_): *δ* 171.9, 165.8, 163.1, 151.1, 141.2, 137.2, 131.9, 131.2, 130.4, 129.3, 129.3, 129.3, 129.2, 125.6, 125.6, 125.4, 122.3, 121.2, 119.7, 119.7, 111.0, 109.7, 105.8, 38.2, 20.7, 20.7. HREI-MS: *m*/*z*, calcd for C_26_H_23_N_7_S_2_ [M]^+^ 497.1456; found: 497.1440.

*2-(2-((5H-[1,2,4]triazino[5,6-b]indol-3-yl)thio)-1-(4-nitrophenyl)ethylidene)-N-phenylhydrazine-1-carbothioamide* (**9**) Yield: 71%; ^1^H-NMR: (500 MHz, DMSO-*d*_6_): *δ* 13.34 (s, 1H, NH), 11.25 (s, 2H, NH), 7.79 (d, *J* = 6.75 Hz, 3H, Ar), 7.54 (d, *J* = 5.8 Hz, 3H, Ar), 7.36 (ddd, *J* = 0.9, 6.4 Hz, 1H, Ar), 7.28 (dd, *J* = 6.55 Hz, 3H, Ar), 7.11 (m, 1H, Ar), 6.97 (d, *J* = 6.5 Hz, 2H, Ar), 5.1 (s, 2H, CH_2_). ^13^C-NMR (125 MHz, DMSO-*d*_6_): *δ* 171.9, 165.8, 163.1, 151.1, 141.2, 137.2, 131.9, 131.2, 130.4, 130.4, 130.4, 129.3, 129.3, 129.3, 129.2, 125.6, 125.4, 122.3, 121.2, 119.7, 119.7, 111.0 105.8, 38.2. HREI-MS: *m*/*z*, calcd for C_24_H_18_N_8_O_2_S_2_ [M]^+^ 514.0994; found: 514.0980.

*2-(2-((5H-[1,2,4]triazino[5,6-b]indol-3-yl)thio)-1-(4-methoxyphenyl)ethylidene)-N-(2-bromophenyl)hydrazine-1-carbothioamide* (**10**) Yield: 71%; ^1^H-NMR: (500 MHz, DMSO-*d*_6_): *δ* 13.30 (s, 1H, NH), 11.25 (s, 1H, NH), 10.86 (s, 1H, NH), 7.84 (t, *J* = 7.2 Hz, 2H, Ar), 7.54 (d, *J* = 6.2 Hz, 2H, Ar), 7.35 (t, *J* = 6.35, 2H, Ar), 7.10 (t, *J* = 6.55, 2H, Ar), 7.03 (d, *J* = 7.1 Hz, 2H, Ar), 6.99 (m, 2H, Ar), 4.80 (s, 2H, CH_2_), 3.81 (s, 3H, OCH_3_). ^13^C-NMR (125 MHz, DMSO-*d*_6_): *δ* 171.9, 165.8, 163.1, 151.1, 141.2, 137.2, 131.9, 131.2, 130.4, 130.4, 130.4, 129.3, 129.2, 125.6, 125.6, 125.4, 122.3, 121.2, 119.7, 119.7, 111.0, 109.7, 105.8, 38.2, 55.2. HREI-MS: *m*/*z*, calcd for C_25_H_20_BrN_7_OS_2_ [M]^+^ 577.0354; found: 577.0342.

*2-(2-((5H-[1,2,4]triazino[5,6-b]indol-3-yl)thio)-1-(3-nitrophenyl)ethylidene)-N-(2-bromophenyl)hydrazine-1-carbothioamide* (**11**) Yield: 70%; ^1^H-NMR: (500 MHz, DMSO-*d*_6_): *δ* 13.30 (s, 1H, NH), 11.25 (s, 2H, NH), 8.71 (m, 2H, Ar), 8.37 (d, *J* = 6.5 Hz, 1H, Ar), 8.33 (d, *J* = 6.35, 1H, Ar), 8.21 (m, 2H, Ar), 7.95 (s, 1H, Ar), 7.75 (t, *J* = 6.6 Hz, 1H, Ar), 7.55 (d, *J* = 6.25 Hz, 1H, Ar), 7.36 (t, *J* = 6.15 Hz, 1H, Ar), 7.11 (t, *J* = 6.2, 1H, Ar), 6.9 (d, *J* = 6.5, 1H, Ar), 5.5 (s, 2H, CH_2_). ^13^C-NMR (125 MHz, DMSO-*d*_6_): *δ* 171.9, 165.8, 163.1, 151.1, 141.2, 137.2, 131.9, 131.2, 130.4, 130.4, 130.4, 129.3, 129.3, 129.3, 129.2, 125.6, 125.4, 123.3, 121.5, 119.7, 119.7, 111.0 105.8, 38.2. HREI-MS: *m*/*z*, calcd for C_24_H_17_BrN_8_O_2_S_2_ [M]^+^ 592.0099; found: 592.0084.

*2-(2-((5H-[1,2,4]triazino[5,6-b]indol-3-yl)thio)-1-(3-nitrophenyl)ethylidene)-N-(p-tolyl)hydrazine-1-carbothioamide* (**12**) Yield: 70%; ^1^H-NMR: (500 MHz, DMSO-*d*_6_): *δ* 11.25 (s, 2H, NH), 10.86 (s, 1H, NH), 7.80 (m, 3H, Ar), 7.68 (s, 1H, Ar), 7.54 (d, *J* = 4.8 Hz, 2H, Ar), 7.35 (t, *J* = 6.4, 1H, Ar), 7.25 (m, 3H, Ar), 7.11 (t, *J* = 6.3, 1H, Ar), 6.98 (m, 1H, Ar), 4.8 (s, 2H, CH_2_), 1.9 (s, 3H, CH_3_). ^13^C-NMR (125 MHz, DMSO-*d*_6_): *δ* 171.9, 165.8, 163.1, 151.1, 141.2, 137.2, 131.9, 131.2, 130.4, 130.4, 130.4, 129.3, 129.3, 129.3, 129.2, 125.6, 125.4, 122.3, 121, 119.7, 119.7, 111.0 105.8, 38.2, 21.0. HREI-MS: *m*/*z*, calcd for C_25_H_20_N_8_O_2_S_2_ [M]^+^ 528.1151; found: 528.1136.

*2-(2-((5H-[1,2,4]triazino[5,6-b]indol-3-yl)thio)-1-([1,1′-biphenyl]-4-yl)ethylidene)-N-(2,6-dimethylphenyl)hydrazine-1-carbothioamide* (**13**) Yield: 64%; ^1^H-NMR: (500 MHz, DMSO-*d*_6_): *δ* 11.25 (s, 2H, NH), 10.86 (s, 1H, NH), 8.0 (d, *J* = 7 Hz, 1H, Ar), 7.9 (d, *J* = 7 Hz, 1H, Ar), 7.7 (d, *J* = 6.2 Hz, 1H, Ar), 7.5 (d, *J* = 7, 2H, Ar), 7.4 (d, *J* = 7.1 Hz, 2H, Ar), 7.36 (m, 3H, Ar), 7.34 (d, *J* = 6.25 Hz, 2H, Ar), 7.21 (d, *J* = 7.3 Hz, 1H, Ar), 7.17 (d, *J* = 7 Hz, 1H, Ar), 7 (t, *J* = 6.9 Hz, 1H, Ar), 6.9 (d, *J* = 6.5 Hz, 1H, Ar), 4.9 (s, 2H, CH_2_), 1.88 (s, 6H, CH_3_). ^13^C-NMR (125 MHz, DMSO-*d*_6_): *δ* 192.8, 166.2, 163.1, 146.4, 144.8, 141, 139.5, 139.4, 138.8, 134.7, 133.5, 129.0, 129.0, 128.9, 128.4, 127.5, 127.0, 126.9, 126.8, 126.4, 126.2,122.4, 122.3, 121.4, 119.8, 117.5, 112.6, 111.0, 106.9, 38.2, 21.0, 21.0. HREI-MS: *m*/*z*, calcd for C_32_H_27_N_7_S_2_ [M]^+^ 573.1769; found: 573.1755.

*2-(2-((5H-[1,2,4]triazino[5,6-b]indol-3-yl)thio)-1-(3-nitrophenyl)ethylidene)-N-(2,6-dimethylphenyl)hydrazine-1-carbothioamide* (**14**) Yield: 70%; ^1^H-NMR: (500 MHz, DMSO-*d*_6_): *δ* 12.57 (s, 2H, NH), 11.28 (s, 1H, NH), 8.36 (t, *J* = 6.5 Hz, 1H, Ar), 8.6(s, 1H, Ar), 8.28 (d, *J* = 6.4, 2H, Ar), 7.96 (m, 3H, Ar), 7.56 (d, *J* = 6.75, 2H, Ar), 7.43 (m, 2H, Ar), 5(s, 2H, CH_2_), 1.91 (s, 6H, CH_3_). ^13^C-NMR (125 MHz, DMSO-*d*_6_): *δ* 171.9, 165.8, 163.1, 151.1, 141.2, 137.2, 131.9, 131.2, 130.4, 130.4, 130.4, 129.3, 129.3, 129.3, 129.2, 125.6, 125.4, 122.3, 121, 119.7, 119.7, 111.0, 105.8, 38.2, 22.1 22.1. HREI-MS: *m*/*z*, calcd for C_26_H_22_N_8_O_2_S_2_ [M]^+^ 542.1307; found: 542.1293.

*2-(2-((5H-[1,2,4]triazino[5,6-b]indol-3-yl)thio)-1-(4-methoxyphenyl)ethylidene)-N-(2,6-dimethylphenyl)hydrazine-1-carbothioamide* (**15**) Yield: 75%; ^1^H-NMR: (500 MHz, DMSO-*d*_6_): *δ* 11.25 (s, 2H, NH), 10.86 (s, 1H, NH), 7.84 (d, *J* = 7.25 Hz, 3H, Ar), 7.54 (d, *J* = 6.2 Hz, 1H, Ar), 7.34 (d, *J* = 6.5 Hz, 1H, Ar), 7.10 (t, *J* = 6.3, 1H, Ar), 6.99 (t, *J* = 7.2 Hz, 5H, Ar), 4.4 (s, 2H, CH_2_), 3.81 (s, 3H, OCH_3_), 2.3 (s, 6H, CH_3_). ^13^C-NMR (125 MHz, DMSO-*d*_6_): *δ* 165.8, 163.1, 159.0, 150.9, 141.2, 131.8, 130.3, 127.0, 127.0, 127.0, 126.9, 126.7, 122.3, 122.3, 121.0, 119.7, 119.7,114.1, 114.1, 114.1, 114.0, 111.0,104.6, 55.2, 38.2, 22.2, 22.2. HREI-MS: *m*/*z*, calcd for C_27_H_25_N_7_OS_2_ [M]^+^ 527.1562; found: 527.1551.

*2-(2-((5H-[1,2,4]triazino[5,6-b]indol-3-yl)thio)-1-(4-methoxyphenyl)ethylidene)-N-(4-fluorophenyl)hydrazine-1-carbothioamide* (**16**) Yield: 74%; ^1^H-NMR: (500 MHz, DMSO-*d*_6_): *δ* 12.5 (s, 2H, NH), 10.86 (s, 1H, NH), 8.28 (d, *J* = 6.5 Hz, 1H, Ar), 7.90 (d, *J* = 7.3 Hz, 2H, Ar), 7.66 (m, 1H, Ar), 7.55 (d, *J* = 6.75 Hz, 1H, Ar), 7.43 (d, *J* = 6.4 Hz, 2H, Ar), 7.39 (m, 1H, Ar), 7.08 (d, *J* = 7.3 Hz, 2H, Ar), 7.02 (d, *J* = 6.75 Hz, 2H, Ar), 4.94 (s, 2H, CH_2_), 3.46 (s, 3H, OCH_3_). ^13^C-NMR (125 MHz, DMSO-*d*_6_): *δ* 191.5, 166.3, 163.4, 163.3 146.7, 146.3, 140.9, 140.2, 131.5, 131.1, 130.8, 130.6, 129.2, 128.7, 128.6, 122.4, 122.2, 121.7, 121.3, 117.5, 114.0, 113.9, 112.6, 55.5, 38.0. HREI-MS: *m*/*z*, calcd for C_25_H_20_FN_7_OS_2_ [M]^+^ 517.1155; found: 517.1143.

*2-(2-((5H-[1,2,4]triazino[5,6-b]indol-3-yl)thio)-1-(3-nitrophenyl)ethylidene)-N-phenylhydrazine-1-carbothioamide* (**17**) Yield: 69%; ^1^H-NMR: (500 MHz, DMSO-*d*_6_): *δ* 11.27 (s, 2H, NH), 10.86 (s, 1H, NH), 8.6 (s, 1H, Ar), 8.36 (d, *J* = 6.5, 3H, Ar), 8.19 (m, 1H, Ar), 7.9(d, *J* = 5.2, 1H, Ar), 7.75 (m, 2H, Ar), 7.55 (d, *J* = 6.15 Hz, 2H, Ar), 7.36 (t, *J* = 6.35, 1H, Ar), 7.18 (t, *J* = 6.3, 1H, Ar), 6.98 (d, *J* = 6.4, 1H, Ar), 4.9 (s, 2H, CH_2_). ^13^C-NMR (125 MHz, DMSO-*d*_6_): *δ* 171.9, 165.8, 163.1, 151.1, 141.2, 137.2, 131.9, 131.2, 130.4, 130.4, 130.4, 129.3, 129.3, 129.3, 129.2, 125.6, 125.4, 122.3, 121.2, 119.7, 119.7, 111.0, 105.8, 38.2. HREI-MS: *m*/*z*, calcd for C_24_H_18_N_8_O_2_S_2_ [M]^+^ 514.0994; found: 514.0980.

*2-(2-((5H-[1,2,4]triazino[5,6-b]indol-3-yl)thio)-1-(4-methoxyphenyl) ethylidene)-N-(3,4-dichlorophenyl)hydrazine-1-carbothioamide* (**18**) Yield: 71%; ^1^H-NMR: (500 MHz, DMSO-*d*_6_): *δ* 12.6 (s, 2H, NH), 11.25 (s, 1H, NH), 8.28 (d, *J* = 6.45 Hz, 1H, Ar), 8.10 (d, *J* = 7.3 Hz, 2H, Ar), 7.69 (m, 2H, Ar), 7.61 (s, 1H, Ar), 7.55 (d, *J* = 6.7 Hz, 1H, Ar), 7.43 (t, *J* = 6.3 Hz, 2H, Ar), 7.12 (d, *J* = 7.35 Hz, 2H, Ar), 4.9 (s, 2H, CH_2_), 3.81(s, 3H, OCH_3_). ^13^C-NMR (125 MHz, DMSO-*d*_6_): *δ* 171.9, 165.8, 163.1, 159, 150.9, 141.2, 131.8, 131.8, 130.3, 128.2, 127, 126.9, 126.7, 123, 121.3, 121.2, 119.7, 119.7, 118.3 114.1, 114, 111, 104, 55.2, 38.2. HREI-MS: *m*/*z*, calcd for C_25_H_19_Cl_2_N_7_OS_2_ [M]^+^ 567.0470; found: 567.0458.

*2-(2-((5H-[1,2,4]triazino[5,6-b]indol-3-yl)thio)-1-([1,1′-biphenyl]-4-yl)ethylidene)-N-(3,4-dichlorophenyl)hydrazine-1-carbothioamide* (**19**) Yield: 77%; ^1^H-NMR: (500 MHz, DMSO-*d*_6_): *δ* 12.5 (s, 2H, NH), 11.3 (s, 1H, NH), 8.27 (d, *J* = 6.5 Hz, 1H, Ar), 8.21 (d, *J* = 6.95 Hz, 2H, Ar), 7.91 (d, *J* = 6.9 Hz, 2H, Ar), 7.80 (d, *J* = 6.2 Hz, 2H, Ar), 7.69 (d, *J* = 7Hz, 2H, Ar), 7.55 (m, 4H, Ar), 7.47(m, 3H, Ar), 5.03 (s, 2H, CH_2_). ^13^C-NMR (125 MHz, DMSO-*d*_6_): *δ* 192.8, 166.2, 163.1, 146.4, 144.8, 141, 139.5, 139.4, 138.8, 134.7, 133, 129.0, 129.0, 128.9, 128.4, 127.5, 127.0, 126.9, 126.8, 126.4, 126.2,122.6, 122.3, 121.4, 119.8, 117.5, 112.6, 111.0, 106.9, 38.2. HREI-MS: *m*/*z*, calcd for C_30_H_21_Cl_2_N_7_S_2_ [M]^+^ 613.0677; found: 613.0664.

*2-(2-((5H-[1,2,4]triazino[5,6-b]indol-3-yl)thio)-1-(p-tolyl)ethylidene)-N-(2,6-dimethylphenyl)hydrazine-1-carbothioamide* (**20**) Yield: 68%; ^1^H-NMR: (500 MHz, DMSO-*d*_6_): *δ* 11.25 (s, 2H, NH), 10.86 (s, 1H, NH), 7.79 (d, *J* = 6.6 Hz, 2H, Ar), 7.53 (d, *J* = 4.1 Hz, 3H, Ar), 7.35 (t, *J* = 6.45 Hz, 1H, Ar), 7.23 (d, *J* = 6.5 Hz, 3H, Ar), 7.10 (t, *J* = 6.3 Hz, 1H, Ar), 6.97 (d, *J* = 6.5, 1H, Ar), 2.3(s, 6H, CH_3_), 4.8 (s, 2H, CH_2_), 1.91 (s, 3H, CH_3_). ^13^C-NMR (125 MHz, DMSO-*d*_6_): *δ* 171.9, 165.8, 163.1, 151.1, 141.2, 137.2, 131.9, 131.2, 130.4, 129.3, 129.3, 129.2, 129.2, 125.6, 125.6, 125.4, 122.3, 121.2, 119.7, 119.7, 111.0, 109.7, 105.8, 38.2, 21.0, 21.0, 20.7. HREI-MS: *m*/*z*, calcd for C_27_H_25_N_7_S_2_ [M]^+^ 511.1613; found: 511.1600.

*2-(2-((5H-[1,2,4]triazino[5,6-b]indol-3-yl)thio)-1-(p-tolyl)ethylidene)-N-(3,4-dichlorophenyl)hydrazine-1-carbothioamide* (**21**) Yield: 68%; ^1^H-NMR: (500 MHz, DMSO-*d*_6_): *δ* 11.25 (s, 2H, NH), 10.86 (s, 1H, NH), 7.77 (d, *J* = 6.4 Hz, 2H, Ar), 7.54 (d, *J* = 4.8 Hz, 2H, Ar), 7.32 (t, *J* = 6.2 Hz, 1H, Ar), 7.23 (d, *J* = 6.5 Hz, 3H, Ar), 7.10 (t, *J* = 6.25 Hz, 2H, Ar), 6.97 (s, 1H, Ar), 4.9 (s, 2H, CH_2_), 1.91 (s, 3H, CH_3_). ^13^C-NMR (125 MHz, DMSO-*d*_6_): *δ* 171.9, 165.8, 163.1, 159, 150.9, 141.2,131.8, 131.8, 130.3, 128.2, 127, 126.9, 126.7, 123, 121.3, 121.2, 119.7, 119.7, 118.3 114.1, 114, 111, 104, 38.2, 20.1. HREI-MS: *m*/*z*, calcd for C_25_H_19_Cl_2_N_7_S_2_ [M]^+^ 551.0520; found: 551.0510.

*2-(2-((5H-[1,2,4]triazino[5,6-b]indol-3-yl)thio)-1-(4-methoxyphenyl)ethylidene)-N-(2,3-dimethylphenyl)hydrazine-1-carbothioamide* (**22**) Yield: 62%; ^1^H-NMR: (500 MHz, DMSO-*d*_6_): *δ* 11.25 (s, 2H, NH), 10.86 (s, 1H, NH), 7.79 (d, *J* = 6.6 Hz, 2H, Ar), 7.51 (d, *J* = 4.5 Hz, 3H, Ar), 7.30 (t, *J* = 6.45 Hz, 1H, Ar), 7.20 (d, *J* = 6.7 Hz, 3H, Ar), 7.10 (t, *J* = 6.3 Hz, 1H, Ar), 6.93 (d, *J* = 6.5, 1H, Ar), 4.8 (s, 2H, CH_2_), 3.8 (s, 3H, OCH_3_), 2.3 (s, 6H, CH_3_). ^13^C-NMR (125 MHz, DMSO-*d*_6_): *δ* 171.9, 165.8, 163.1, 151.1, 141.2, 137.2, 131.9, 131.2, 130.4, 129.3, 129.3, 129.3, 129.2, 125.6, 125.6, 125.4, 122.3, 121.2, 119.7, 119.7, 111.0, 109.7, 105.8, 55.2, 38.2, 21.0, 21.0. HREI-MS: *m*/*z*, calcd for C_27_H_25_N_7_OS_2_ [M]^+^ 527.1562; found: 527.1551.

*2-(2-((5H-[1,2,4]triazino[5,6-b]indol-3-yl)thio)-1-(3-nitrophenyl)ethylidene)-N-(2,3-dichlorophenyl)hydrazine-1-carbothioamide* (**23**) Yield: 65%; ^1^H-NMR (500 MHz, DMSO-*d*_6_): *δ* 11.29 (s, 2H, NH), 10.86 (s, 1H, NH), 8.72 (d, *J* = 6.4 Hz, 1H, Ar), 8.37 (m, 1H, Ar), 8.23 (d, *J* = 6.2, 1H, Ar), 7.96 (s, 1H, Ar), 7.75 (m, 3H, Ar), 7.56 (d, *J* = 6.25, 1H, Ar), 7.37 (t, *J* = 6.25, 1H, Ar), 7.12 (t, *J* = 6.25, 1H, Ar), 6.98 (d, *J* = 6.5, 1H, Ar), 4.8 (s, 2H, CH_2_). ^13^C-NMR (125 MHz, DMSO-*d*_6_): *δ* 171.9, 165.8, 163.1, 159, 150.9, 141.2, 131.8, 131.8, 130.3, 129.2, 127, 126.9, 126.7, 123, 122.3, 121.2, 119.7, 119.7, 118.3 114.1, 114, 111, 104, 38.2. HREI-MS: *m*/*z*, calcd for C_24_H_16_Cl_2_N_8_O_2_S_2_ [M]^+^ 582.0215; found: 582.0202.

*2-(2-((5H-[1,2,4]triazino[5,6-b]indol-3-yl)thio)-1-([1,1′-biphenyl]-4-yl)ethylidene)-N-phenylhydrazine-1-carbothioamide* (**24**) Yield: 74%; ^1^H-NMR (500 MHz, DMSO-*d*_6_): *δ* 11.27 (s, 2H, NH), 10.86 (s, 1H, NH), 8.2 (d, *J* = 6.8 Hz, 2H, Ar), 7.75 (m, 7H, Ar), 7.55 (d, *J* = 6.2 Hz, 1H, Ar), 7.49 (t, *J* = 6.3 Hz, 3H, Ar), 7.39 (m, 3H, Ar), 7.19 (t, *J* = 6.25 Hz, 1H, Ar), 6.98 (d, *J* = 6.45 Hz, 1H, Ar), 5.03 (s, 2H, CH_2_). ^13^C-NMR (125 MHz, DMSO-*d*_6_): *δ* 192.8, 166.2, 163.1, 146.4, 144.8, 141, 139.5, 139.4, 138.8, 134.7, 133, 129.0, 129.0, 128.9, 128.4, 127.5, 127.0, 126.8, 126.8, 126.4, 126.2, 122.4, 122.3, 121.4, 119.8, 117.5, 112.6, 111.0, 106.9, 38.2. HREI-MS: *m*/*z*, calcd for C_30_H_23_N_7_S_2_ [M]^+^ 545.1456; found: 545.1439.

*2-(2-((5H-[1,2,4]triazino[5,6-b]indol-3-yl)thio)-1-(4-methoxyphenyl)ethylidene)-N-phenylhydrazine-1-carbothioamide* (**25**) Yield: 73%; ^1^H-NMR (500 MHz, DMSO-*d*_6_): *δ* 11.25 (s, 2H, NH), 10.6 (s, 1H, NH), 7.84 (d, *J* = 7.3 Hz, 3H, Ar), 7.54 (d, *J* = 6.25 Hz, 1H, Ar) 7.4 (m, 1H, Ar), 7.36 (t, *J* = 6.4 Hz, 1H, Ar), 7.11 (t, *J* = 6.75 Hz, 2H, Ar), 6.99 (t, *J* = 7.1 Hz, 5H, Ar), 4.9 (s, 2H, CH_2_), 3.84 (s, 3H, OCH_3_). ^13^C-NMR (125 MHz, DMSO-*d*_6_): *δ* 171.9, 165.8, 163.1, 159.0,150.9, 141.2, 131.8, 130.4, 127.0, 126.9, 126.9, 126.9, 126.7, 122.3, 121.2, 119.7, 119.7,114.2, 114.0, 114.0, 114.0, 111.0, 104.6, 55.2, 38.2. HREI-MS: *m*/*z*, calcd for C_25_H_21_N_7_OS_2_ [M]^+^ 499.1249; found: 499.1233.

### 4.2. α-Glucosidase Assay Protocol

The α-glucosidase activity was executed according to Fazal et al. [31]. The following chemicals were used (concentrations):70 μL of 50 mM phosphate buffer (pH 6.8);10 μL (0.5 mM in methanol) test compounds; and10 μL (0.057 units, Sigma Inc.) of enzyme solution in buffer.
For details on the experiment, kindly see Reference [31].

### 4.3. Molecular Docking

Molecular docking was performed on the active compounds to identify possible binding modes that explained the reason for their potency. The method used for molecular docking was as mentioned in our previous paper with slight modifications [32]. The molecular docking study was conducted using a homology model for α-glucosidase. The structures of all compounds were prepared using Chem3D by CambridgeSoft. The geometry and energy of the structures were optimized using MMFF94. GOLD was used to identify the binding modes of the active compounds responsible for the activity. The Chemscore fitness function with default settings was employed in this study. The protein sequence for Baker’s yeast α-glucosidase (MAL12) was obtained from uniprot (http://www.uniprot.org). A homology model for *Saccharomyces. cerevisiae* glucosidase was built using the crystal structure of α-d-glucose-bound isomaltase from *S. cerevisiae* (PDB ID: 3A4A), which shares a 72% identical and 85% similar sequence to α-glucosidase. The sequence alignment and homology modeling were performed using Swiss-Model, which is a fully automated homology modeling pipeline (SWISS-MODEL) managed by the Swiss Institute of Bioinformatics. The docking results were visualized using Discovery Studio visualizer 3.5 and PyMol. The homology model was evaluated using PROCHECK.

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
