# Peer review of "Synthesis of Novel Triazinoindole-Based Thiourea Hybrid: A Study on α-Glucosidase Inhibitors and Their Molecular Docking"

_molecules, 2019, doi:10.3390/molecules24213819_

Round 1

Reviewer 1 Report

In the manuscript, the authors describe preparation of 1-25 and their a-glucosidase inhibitory potential. The biologically activity was discussed using result of docking study. The manuscript may be acceptable to be published in Molecules after addressing the minor points below.

In Table 1, the R substitutent of the structures of 3, 4, 6, 13, 19, and 24 should be revised to biphenyl instead of para-methylbiphenyl.

In experimental section, 1H NMR data of 2, It is difficult to understand the signal of 6.98 (t, J = 7.5, 3H, Ar).

Author Response

1# In Table 1, the R substitutent of the structures of 3, 4, 6, 13, 19, and 24 should be revised to biphenyl instead of para-methylbiphenyl.

 Reply: Corrected according to the kind reviewer suggestion.

2# In experimental section, 1H NMR data of 3, It is difficult to understand the signal of 6.98 (t, J = 7.5, 3H, Ar).

 Reply: Now the NMR data of compound 3 is corrected according to the kind reviewer suggestion.

Reviewer 2 Report

The abstract does not summarize the results, discussion and conclusion, just methodology. Discussion about obtained results is missing. Authors should relate the results with previous studies. How did authors validate the results of molecular docking? Energy-based scoring function of molecular docking is missing. Computational method is missing in Material and methods.

Manuscript is half-done and only after proper completion could be considered for publication.

Author Response

Reviewer 2:

1#. The abstract does not summarize the results, discussion and conclusion, just methodology. Discussion about obtained results is missing. Authors should relate the results with previous studies.

Reply: Corrected the abstract now and we have related our current study with the previous study in the introduction part already.

2#. How did authors validate the results of molecular docking?

Reply: Validation had been performed as reported in our previous paper [Imran, S., Taha, M., Ismail, N.H., Kashif, S.M., Rahim, F., Jamil, W., Hariono, M., Yusuf, M. and Wahab, H., 2015. Synthesis of novel flavone hydrazones: in-vitro evaluation of α-glucosidase inhibition, QSAR analysis and docking studies. European journal of medicinal chemistry, 105, pp.156-170] and the validation steps had been included as recommended by worthy reviewer suggestion.

3#. Energy-based scoring function of molecular docking is missing.

Reply: Molecular docking had been performed using chemscore fitness function which is an empirical scoring function and it has been added to the manuscript as suggested by worthy reviewer.

Computational method is missing in Material and methods.

Reply: Methodology for molecular docking had been included as suggested by respected reviewer.

Reviewer 3 Report

Synthesis item is well described. The order of the compounds in the table should be change to allow comparison of the scaffolds properly. 

The conclusions arrived in the "biological activity" section should be supported by a protein-ligand complex analysis. The authors should include some activity data/plots (i.e enzyme kinetics) in this section.

"Docking study" section images should be made with some molecule visualization program (i.e VMD, pymol) the and complexes scheme's should be included in supplementary information.   

The authors should include a methodology description (program, parameters, etc) for docking simulation.

Author Response

1#. Synthesis item is well described. The order of the compounds in the table should be changed to allow comparison of the scaffolds properly.

Reply: The order of the table is changed according to the kind reviewer suggestion.

2#. The conclusions arrived in the "biological activity" section should be supported by a protein-ligand complex analysis. The authors should include some activity data/plots (i.e enzyme kinetics) in this section.

Reply: We have revised molecular docking part but as far as enzyme kinetics study concerned we do not have capability to performed such experiments.

3#. "Docking study" section images should be made with some molecule visualization program (i.e VMD, pymol) and complexes schemes should be included in supplementary information.

Reply: Molecular docking study images had been prepared using pymol as recommended by worthy reviewer.

4#. The authors should include a methodology description (program, parameters, etc) for docking simulation.

Reply: docking simulation method had been included in the methodology section as suggested by respected   reviewer.

Round 2

Reviewer 2 Report

Second review Taha

The abstract still does not summarize the results, discussion and conclusion, just methodology. Instructions for Authors: „The abstract should be a single paragraph and should follow the style of structured abstracts, but without headings: 1) Background: Place the question addressed in a broad context and highlight the purpose of the study; 2) Methods: Describe briefly the main methods or treatments applied. Include any relevant preregistration numbers, and species and strains of any animals used. 3) Results: Summarize the article's main findings; and 4) Conclusion: Indicate the main conclusions or interpretations.

Values of mentioned Chemscore fitness function were still not given in the manuscript.

Athors said: „Acarbose was docked into α-glucosidase from Sugar beet (PDB code: 3W37) and compared by 106 superimposing with the native ligand in the protein as mentioned in Imran et. al. (2016) [24].“, but further discussion is still missing.

Other:

Line 106: Sugar should be sugar

Author Response

Reviewer 1

The abstract still does not summarize the results, discussion and conclusion, just methodology. Instructions for Authors: „The abstract should be a single paragraph and should follow the style of structured abstracts, but without headings: 1) Background: Place the question addressed in a broad context and highlight the purpose of the study; 2) Methods: Describe briefly the main methods or treatments applied. Include any relevant preregistration numbers, and species and strains of any animals used. 3) Results: Summarize the article's main findings; and 4) Conclusion: Indicate the main conclusions or interpretations.

Reply: we have revised the manuscript as suggested by respected reviewer

Values of mentioned Chemscore fitness function were still not given in the manuscript.

Reply: The chemscore values had been added as suggested by worthy reviewer.

Athors said: „Acarbose was docked into α-glucosidase from Sugar beet (PDB code: 3W37) and compared by 106 superimposing with the native ligand in the protein as mentioned in Imran et. al. (2016) [24].“, but further discussion is still missing.

Reply: Details on validation process through co-crystal ligand redocking had been included in the discussion as suggestion by respected reviewer.

Other:

Line 106: Sugar should be sugar

Reply: Change had been made accordingly as suggested by worthy reviewer.

Reviewer 3 Report

The description of the figures should be improved in the epigraph.

The authors didn't include kinetic curves even though the present IC50 values for each compound. I strongly recommend a graph with this information.

Author Response

Reviewer 2

The description of the figures should be improved in the epigraph.

Reply: Figure description had been improved as suggested by respected reviewer.

The authors didn't include kinetic curves even though the present IC50 values for each compound. I strongly recommend a graph with this information.

Reply: Our work is collaborative work bioassay was performed by our co-worker nowadays he is on leave need to wait for two to three weeks to get all kinetic curves.